# Comparison of Current Approaches to Lemmatization: A Case Study in Estonian

**Aleksei Dorkin**
Institute of Computer Science
University of Tartu
`aleksei.dorkin@ut.ee`

**Kairit Sirts**
Institute of Computer Science
University of Tartu
`kairit.sirts@ut.ee`

## Abstract

This study evaluates three different lemmatization approaches to Estonian—Generative character-level models, Pattern-based word-level classification models, and rule-based morphological analysis. According to our experiments, a significantly smaller Generative model consistently outperforms the Pattern-based classification model based on EstBERT. Additionally, we observe a relatively small overlap in errors made by all three models, indicating that an ensemble of different approaches could lead to improvements.

## 1 Introduction

Recently, two different approaches have been adopted for model-based lemmatization. The Generative approach is based on encoder-decoder models and they generate the lemma character by character conditioned on the word form with its relevant context (Qi et al., 2020; Bergmanis and Goldwater, 2018). The Pattern-based approach treats lemmatization as a classification task (Straka, 2018), where each class is a transformation rule. When the correct rule is applied to a word-form, it unambiguously transforms the word-form to its lemma.

Our aim in this paper is to compare the performance of these two lemmatization approaches in Estonian. As a third approach, we also adopt the Estonian rule-based lemmatizer Vabamorf (Kaalep and Vaino, 2001). As all three approaches rely on different formalisms to lemmatization, we are also interested in the complementarity of these methods. The Generative approach is the most flexible, it has the largest search space and therefore it can occasionally result in hallucinating non-existing morphological transformations. On the other hand, the search space of the Pattern-based approach is much smaller as the model only has to correctly

choose a single transformation class. However, if the required transformation is not present in the set of classes then the model is blocked from making the correct prediction. Similarly, the rule-based system can be highly precise but if it encounters a word that is absent from its dictionary the system can be clueless even if this word is morphologically highly regular.

One problem with the recently proposed pattern-based approach implemented in the UDPipe2 is that the transformation rules mix the casing and morphological transformations. This means that for many morphological transformations there will be two rules in the ruleset—one for the lower-cased version of the word and another for the same word with the capital initial letter that needs to be lowered for the lemma—which increases the size of the ruleset considerably and thus artificially complicates the prediction task. Thus, in many cases a more optimal approach would be to treat casing separately from the lemmatization. Additionally, in the UD Estonian treebanks, lemmas include annotations of derivational and compounding processes marked by special symbols. However, these annotations are inconsistent in the data which confuses the models and also complicates the transformation rules. Thus, for our evaluation to be unaffected by these factors, we also train our models on the lowercased data with the special symbols removed.

Lemmatization models are commonly *token-based*, meaning that if the same word-form (with its relevant context) appears several times in the dataset, these repeating instances are kept in the data and thus the training and evaluation sets reflect the natural distribution of words. In contrast, for the morphological reinflection task, the custom has been to train *type-based* models, in which each lemma and morphological feature combination is presented to the model only once. We were interested in how well the type-based approach can work for lemmatization and thus we also experi-

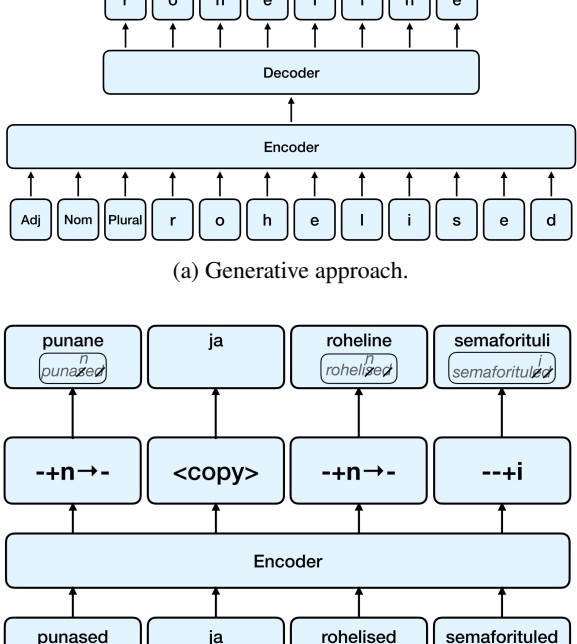

(a) Generative approach.

(b) Pattern-based approach.

Figure 1: Schematic representations of the generative and pattern-based approaches.

mented with type-based models where appropriate.

In sum, our contributions in this paper are first comparing three different lemmatization approaches on two Estonian datasets with different domains with the goal of assessing the complementarity of these systems. Secondly, we investigate the effect of casing and special symbols as well as type- vs token-based training and evaluation for each comparison system.

## 2 Lemmatization approaches

This section gives a brief overview of the current approaches to lemmatization.

### 2.1 Generative approach

Generative lemmatization involves using a neural network to convert a word form, represented as a sequence of characters, into its lemma, also represented as a sequence of characters. The model is trained to predict the lemma in an auto-regressive manner, meaning that it makes predictions one character at a time based on the previously predicted characters. Commonly, generative lemmatization makes use of part of speech and morphological information as context (Qi et al., 2020). However, it is not necessarily limited to that. For example, Bergmanis and Goldwater (2018) propose using

surrounding words, subword units, or characters as context for a given word form.

### 2.2 Pattern-based approach

In the Pattern-based lemmatization, the model assigns a specific transformation class to each word form, and then uses a predetermined rule to transform the word form to the lemma. The approach is not bound to any specific method of classification, or for that matter, representation of input features. For instance, in the UDPipe2 (Straka, 2018), the patterns are sequences of string edit operations, while the Spacy's lemmatizer uses an edit tree structure as a pattern (Müller et al., 2015).

### 2.3 Rule- and lexicon-based approaches

Rule-based approaches to lemmatization use various rule formalisms such as rule cascades or finite state transducers to transform the word form into lemma. For instance, the rule-based machine translation library Apertium also includes rule-based morphological analyzers for many languages (Khanna et al., 2021). For the Estonian language, there is a morphological analyzer called Vabamorf (Kaalep and Vaino, 2001). In the dictionary-based approach, the lemma of a word is determined by looking it up in a special dictionary. The dictionary may include word forms and their POS tags with morphological features, which can be used to identify the correct lemma. Such morphologial dictionaries include for instance Unimorph (McCarthy et al., 2020) and UD Lexicons (Sagot, 2018).

What these approaches have in common is that, intrinsically, they are not able to fully consider the context in which a given word form appears, which prevents them from disambiguating multiple candidates. So, for that purpose they have to rely on separate tools, such as Hidden Markov Models. They are also language-specific. The advantage, however, is that they are not dependent on the amount of training data, and can be quite precise.

## 3 Data

We use the Estonian Dependency Treebank (EDT) and the Estonian Web Treebank (EWT) from the Universal Dependencies collection version 2.10. The EDT comprises several genres such as newspaper texts, fiction, scientific articles, while the EWT is composed of texts from internet blogs and forums. The statistics of both datasets are given in Table 1.

|  | train | dev | test |
|---|---|---|---|
| EDT # of sentences | 24633 | 3125 | 3214 |
| EDT # of tokens | 344953 | 44686 | 48532 |
| EWT # of sentences | 4579 | 833 | 913 |
| EWT # of tokens | 55143 | 10012 | 13176 |

Table 1: Number of sentences and tokens per split in Estonian Dependency Treebank and Estonian Web Treebank as of version 2.10.

## 4 Implementation

For the Generative approach, we adopted the neural transducer by Wu et al. (2020), previously used for morphological reinflection. Neural transducer is a character-level transformer, which takes individual characters of a word form and morphological tags as input, and outputs the resulting lemma character-by-character.

For the Pattern-based model we adopted an approach similar to UDpipe2 (Straka, 2018). We used a transformer-based token classification model by fine-tuning EstBERT (Tanvir et al., 2020) to predict the correct transformation class (form → lemma) for every token in a sentence. Our model uses HuggingFace (Wolf et al., 2020) TokenClassification implementation. Moreover, we reuse the code to generate transformation classes from UDpipe2.[1]

For the rule-based approach, we adopted the Estonian rule-based morphological analyzer Vabamorf (Kaalep and Vaino, 2001). We used Vabamorf via EstNLTK, which is a library that provides an API to various Estonian language technology tools (Orasmaa et al., 2016). We utilized Vabamorf's HMM-based disambiguation capabilities to output a single lemma for each token.

## 5 Results

Tables 2 and 3 show the results on the EDT and EWT validation sets respectively. Overall, the Generative model (in the token-based training setting, see below) outperforms both the Pattern-based model and the rule-based Vabamorf on both datasets.

The first column (Original) in Table 2 shows results on the EDT data in its original case sensitive form and including special symbols marking derivation and compounding. The second column

[1]https://github.com/ufal/udpipe/blob/udpipe-2/udpipe2_dataset.py

|  | Original | No Sym | Type Eval |
|---|---|---|---|
| Gen Token | 95.49 | 97.59 | 97.61 |
| Gen Type | 91.55 | 95.64 | 95.10 |
| Pattern-based | 95.04 | 96.34 | – |
| Vabamorf | 87.78 | 91.66 | – |
| Vabamorf Oracle | 99.31 | 99.47 | – |

Table 2: Lemmatization accuracy on the EDT validation set. Original: unaltered EDT, No Sym: lowercased EDT with special symbols removed, Type Eval: evaluation on distinct word types with No Sym setting.

| Trained on | EWT | EDT |
|---|---|---|
| Gen Token | 95.88 | 96.28 |
| Gen Type | 94.63 | 95.97 |
| Pattern-based | 95.02 | 87.97 |
| Vabamorf | 91.75 | 91.74 |
| Vabamorf Oracle | 96.98 | 96.98 |

Table 3: Lemmatization accuracy on the EWT validation set. The first column contains results for models trained on EWT, the results for models trained on EDT are shown in the second column.

(No Sym) shows the results of models trained on lowercased data with special symbols removed. All approaches show a noticeable improvement in accuracy in the simplified environment. Although the improvement with the Pattern-based model is the smallest, it has the largest implications—ignoring casing and removing special symbols halves the number of transformation classes.

The top part of the Tables 2 and 3 compare the results of the Generative model trained on word tokens and word types. Additionally, the last column of the Table 2 also shows the evaluation on unique types of both the token-based and type-based models trained in the No Sym setting. The Generative model trained on word tokens always performs better than the model trained on unique word types even when evaluated on word types. We conclude that there does not seem to be any disadvantages to token-based training.

In Table 3, EWT validation set is evaluated in two settings. The first column shows the results of the in-domain models trained on the EWT train set, the results in the second column are obtained

with the out-of-domain models trained on the EDT train set. We observe that the Generative models perform well in the cross-domain environment, and outperform the model trained on the EWT train set. Meanwhile, the Pattern-based model trained on the EDT shows a significant drop in performance when evaluated on EWT. Vabamorf also demonstrates a degraded performance on EWT.

The last row in both Tables 2 and 3 show the performance of the Vabamorf in the oracle mode, in which case the prediction is considered correct if the true lemma appears in the list of generated candidates. We observe a significant improvement in the accuracy of Vabamorf in the oracle mode. This means that a large chunk of errors made by the rule-based approach is the result of poor disambiguation, rather than incorrect morphological analysis.

In addition to comparing the performance of different approaches, we are interested in whether there is any complementarity in the errors made by models based on different approaches. Figure 2 presents a Venn diagram of the token-level errors made by each system. We note that the area of the intersection of all three models is relatively small, meaning that the number of words where all models make an error is quite small, suggesting that different approaches can complement each other in an ensemble setting.

## 6 Discussion

Because the UDPipe2's pattern-based approach was highly successful in the Sigmorphon 2019 shared task (Straka et al., 2019), we expected it to perform well also in our case, especially because instead of the frozen BERT weights used in the UDPipe2, we fine-tuned the full model. However, the best shared task lemmatization scores for the Estonian language were obtained with the generative contextual lemmatizer by Bergmanis and Goldwater (2018), which perhaps explains the success of the Generative model also in our experiments. When analyzing the errors made by each three approaches, we can see that the set of errors where all models overlap is relatively small (302 out of 5194, 5.8%), which suggests that different approaches can potentially compensate for each other and thus an ensemble of different methods can be useful.

The rule-based Vabamorf made the largest number of errors. However, when we evaluated it in the oracle mode on EDT, it covered the vast majority of

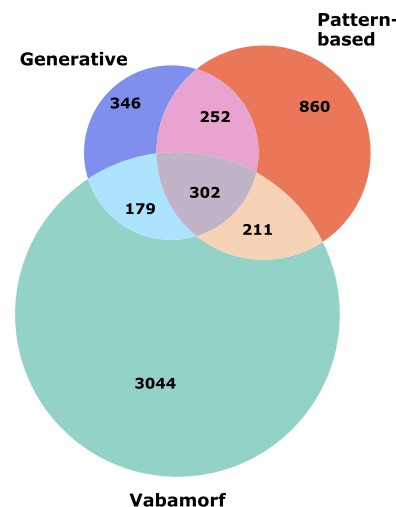

Figure 2: Venn diagram of the token-level lemmatization errors made by each model on the EDT validation set.

correct answers. This implies that Vabamorf could gain a lot from a better disambiguator than the current HMM-based one. This was not the case for EWT which, being a web treebank, contains more word forms (such as neologisms, more recent loanwords, and so on) missing from the Vabamorf's lexicon. Thus, while Vabamorf can be a good solution for formal and grammatically correct Estonian, it is less suitable for more noisy web texts.

The approach to creating transformation rules suggested by the developers of UDpipe may output equivalent rules, i.e., when applying these rules to a surface form, the result is identical. We noticed that the Pattern-based model is able to identify such cases. This means that an incorrectly predicted *label* can result in a correct lemma. For example, two rules ↓0;d¦−−−+m+a and 0;d¦−+m→− transform the third person plural present tense form into the corresponding -ma infinitive (*vabandavad* → *vabandama* "to apologize"). The difference between these rules is that the former rule removes three last letters and adds *ma*-suffix, while the latter removes the last letter, and then replaces the existing letter preceding the existing *a* with *m*. We suggest that such a peculiarity may be used to probe language models for morphological knowledge.

The five most common rules are shown in Table 4. The most common rule is the "do-nothing" rule, which accounts for more than half of the occurrences in the EDT train set. The next three most common rules with smaller but still considerable frequency involve removing suffixes of varying

| %    | Rule        | Description                    |
|------|-------------|--------------------------------|
| 54.1 | ↓0;d¦       | Do nothing                     |
| 8.3  | ↓0;d¦-      | Remove the last letter         |
| 5.2  | ↓0;d¦--     | Remove two last letters        |
| 3.4  | ↓0;d¦---    | Remove three last letters      |
| 3.3  | ↓0;d¦-+m+a  | Replace the last letter with *ma* |

Table 4: Top 5 most common transformation rules present in the train split of the EDT dataset.

length. The last rule fitting into our top-5 list is specific to verbs, replacing the last character with the lemma suffix for verbs. The total set contains a very long tail of transformation rules that appear only a few times or just once, such as rules corresponding to the transformation of infrequently used suppletive forms.

## 7 Conclusion

We compared three lemmatization approaches on two Estonian datasets from different domains and found that on both datasets the Generative encoder-decoder approach trained from scratch outperforms both the rule-based Vabamorf as well as the Pattern-based approach fine-tuned from a large pre-trained language model. We observed complementary error patterns for each three approaches, which suggests that ensembling techniques can take advantage of the complementary strengths of each individual approach.

## Acknowledgments

This research was supported by the Estonian Research Council Grant PSG721.

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
