# OpenReview forum: "Comparison of Current Approaches to Lemmatization: A Case Study in Estonian"
_NoDaLiDa/2023/Conference — NoDaLiDa 2023_

### Official Review · Reviewer_uD4P · 2023-03-07
**The authors make a comparison of three approaches to lemmatization in Estonian and suggest a fourth one but with no implementation.**

**Rating:** 6
**Confidence:** 5

**Review:**

The authors present two neural based approaches and a symbolic one to the automatic lemmatization of Estonian. The two neural based approaches are represented by a Generative encoder-decoder approach trained from scratch and a Pattern-based approach fine-tuned from a large pre-trained language model. The symbolic approach is an adaptation of the Estonian rule-based morphological analyzer Vabamorf, which uses a disambiguator whenever the output is made by multiple choices based on HMM.
The rule-based Vabamorf made the largest number of errors. However, when it was evaluated  in the oracle mode on the corpus EDT, it covered vast majority of correct answers. This was not the case for EWT which, being a web treebank, contains more word forms (such as neologisms, more recent loan-words, and so on) missing from the Vabamorf’s lexicon. Thus, while Vabamorf can be a good solution for formal and grammatically correct Estonian, it is a less suitable for more noisy web texts. However, Vabamorf still achieves best results : 96.98 96.98 for respectively EWT and EDT, and results in the NO SYM(bol) setting are by far the best in the Oracle version, 99.47. The need to have disambiguation is an obvious step in the choice to adopt a rule-based approach, and splitting analysis and disambiguation can be more successful than just relying on a single choice step dependent strictly on the model. Ensembling approaches relying on neural network and probabilistic predictions with symbolic approaches may not be the best solution. I suggest improving rules in the Vabamorf and adding roots and affixes to cover more text, leaving out illegal and highly infrequent word forms that may appear in web scratched texts. Estonian is a highly morphologically rich language and the typology of word construction rules is highly complicated[1] so I assume there might be room for improvement.
As the authors of [2] EstNLTK comment on the performance of neural based lemmatizers, "due to the Zipfian distribution of word lemmas, the neural model needs (possibly infeasibly) large amounts of data to learn correct the lemmatization of rare words... Vabamorf-based lemmatizer, which combines a lexicon with derivation rules, maintains relatively stable performance even on rare words." I agree with the conclusion and the data reported in this second paper that Vabamorf in its latest version is certainly a better choice. Besides the authors themselves blame the generative approach of "... occasionally result in hallucinating non-existing morphological transformations." and the pattern-based approach "...if the required transformation is not present in the set of classes then the model is blocked from making the correct prediction."

[1] Heli Uibo, 2006. Optimizing the finite-state description of Estonian morphology, in Werner (ed.), Proceedings of the 15th NODALIDA conference, Joensuu Ling@JoY 1, pp. 203–209.
[2] Sven Laur, Siim Orasmaa, Dage Sarg, Paul Tammo, 2020. EstNLTK 1.6: Remastered Estonian NLP Pipeline, Proceedings of the 12th Conference on Language Resources and Evaluation (LREC 2020), pages 7152–7160.

PLEASE CORRECT
morphologial --> morphological
strenghts   --->  strengths

**Paper Type:**

Short paper

---

### Official Review · Reviewer_ft82 · 2023-03-07
**Gives insights into useful approaches to lemmatization**

**Rating:** 7
**Confidence:** 5

**Review:**

This paper describes an evaluation of three different lemmatizers applied to Estonian. The results show that a lemmatizer based on a generative model performs the best, and that the three lemmatizers complement each other, indicating that an ensemble system could be a way to move forward.

The comparison is well structured and clearly described, and gives insights into useful approaches to lemmatization.

The authors should go through the text and search for typos. Some typos and minor things found by the reviewer:
ensemble of different approach --> ensemble of different APPROACHES
we also experimented with type-based models were appropriate --> we also experimented with type-based models WHERE appropriate
for each comparison systems --> for each comparison SYSTEM
the models assigns --> the MODEL assigns/the models ASSIGN
several genres such newspaper texts -->  several genres such AS newspaper texts
we EWT validation set is evaluated --> THE EWT validation set is evaluated
it covered vast majority of --> it covered THE vast majority of
it is A less suitable for --> it is less suitable for

**Paper Type:**

Short paper

---

### Decision · Program_Chairs · 2023-03-17

Accept